# Iridium-Catalyzed Synthesis of Chiral 1,2,3-Triazoles Units and Precise Construction of Stereocontrolled Oligomers

**DOI:** 10.3390/molecules28093726

**Published:** 2023-04-26

**Authors:** Xueyan Zhang, Tian Yu, Shengtao Ding

**Affiliations:** 1State Key Laboratory of Organic-Inorganic Composites, College of Chemical Engineering, Beijing University of Chemical Technology, Beijing 100029, China; 2State Key Laboratory of Natural and Biomimetic Drugs, School of Pharmaceutical Sciences, Peking University, Beijing 100191, China

**Keywords:** iridium catalysis, chiral triazole, precise construction, stereoregularity, optical property

## Abstract

Iridium-catalyzed azide-thioalkyne cycloaddition reaction (IrAAC) has proved to be a powerful tool for the synthesis of fully substituted 1,2,3-triazole compounds with exclusive regioselectivity. Here we report its successful use in the precise construction of stereocontrolled oligomers that have great potential in diverse applications. Starting with the azide derived from L-prolinol and different functionalized thioalkynes, chiral 1,2,3-triazole units were fabricated with high efficiency under the IrAAC condition, which were further assembled into stereocontrolled oligotriazoles through metal-free exponential growth strategies. The structure and uniformity of these oligomers were well identified by ^1^H NMR, size-exclusion chromatography, and mass spectrometry, the stereoregularity of which were studied through circular dichroism and circular polarized luminescence analysis.

## 1. Introduction

Iridium-catalyzed transformations have evolved as one significant part in organic synthesis [1,2]. Especially noteworthy is the cycloaddition of unsaturated motifs under iridium catalysis, some of which afford carbocyclic or heterocyclic skeletons in unique manners [3]. For instance, in the metal-catalyzed annulations of alkynes with organic azides, regioselective fabrication of disubstituted 1,2,3-triazoles has been well controlled by different metal complexes [4,5,6], while construction of 1,4,5-trisubstituted 1,2,3-triazoles with exclusive regioselectivity has been hardly achieved [7]. One emblematic example is the iridium-catalyzed cycloaddition of 1-thioalkynes with azide (IrAAC) that can regulate the selectivity in a perfect fashion [8]. In combination with its simple and mild condition, as well as great compatibility with various solvents, IrAAC exhibits great potential in the construction of complex molecules with high tacticity.

Precise assembly of small homochiral molecules into macromolecules is a naturally occurring process in all of the known life-forms, which is also utilized in the production of diverse stereocontrolled peptidomimetics and nucleotidomimetics for various functional applications [9,10,11,12,13]. Construction of abiotic polymers with well-defined monomer sequence and exact chain-length has experienced a significant development in the last two decades [14,15,16,17,18,19]. Nevertheless, efficient introduction of chirality into their backbone is still a challenging issue [20,21,22,23,24,25,26]. One facile protocol is to achieve the preparation of extensible monomers from natural chiral molecules. L-Proline is the only proteinogenic secondary amino acid, the distinctive cyclic structure of which endows its polymer with exceptional conformational rigidity [27,28]. By using one organic azide derived from L-prolinol, we realized the synthesis of chiral fully substituted triazole building blocks and their precise assembly into stereocontrolled sequence-defined oligotriazoles [29]. However, functional variations in the side chain of these oligomers are limited to some extent. By contrast, multifarious side groups were conveniently installed at the C4 position of triazole motifs (Figure 1A) [30]. As a continuation of our research in the precise synthesis of macromolecules [31,32,33,34], here we report the construction of an improved stereocontrolled polytriazole architecture from novel chiral triazole units that are prepared by IrAAC (Figure 1B). To demonstrate the tolerance of this stereoregular skeleton for functionalities, three kinds of oligomers bearing different side chains were precisely fabricated, the structure and monodispersity of which were well characterized by ^1^H NMR, size-exclusion chromatography (SEC) and mass spectrometry (MS) analysis. It is noteworthy that in comparison with the previous work, this newly introduced architecture was established through one metal-free exponential growth strategy at a higher efficiency. The optical properties of the oligomers involving tetraphenylethane (TPE) were further explored by ultraviolet-visible spectroscopy (UV-Vis), fluorescence spectroscopy (FL), circular dichroism (CD), and circularly polarized luminescence (CPL) analysis, the results of which, together with molecular dynamics (MD) simulations, identified their stereoregularity.

## 2. Results and Discussion

### 2.1. Preparation of Chiral Triazole Monomers

One general synthetic route of chiral triazole units is depicted in Figure 2A. The easily obtained bis(2-hydroxyethyl) disulfide with tert-butyldimethylsilyl (TBS) protecting groups was selected to provide extensible termini for internal thioalkynes **1**. Introduction of functionality into the thioalkyne was realized by two different pathways: direct combination of functionalized terminal alkynes with the disulfide or synthesis of thioalkynes with modifiable groups that allow latter installation of functional groups in a more flexible manner. For instance, the phenyl group was introduced by the reaction of phenylacetylene, and the reaction of disulfide with propargyl alcohol afforded the thioalkyne involving a hydroxyl group for the post-introduction of benzyl or tetraphenylethanyl groups. In parallel, the chiral organic azide **2** was prepared from commercially available *N*-Boc-L-prolinol (>99.9% ee) through a simple two-step synthetic protocol, which was well demonstrated in our previous research [29]. Gram-scale coupling of it with different functionalized thioalkynes under the mild and simple IrAAC condition was easily achieved, affording fully substituted triazole motifs **3** in good to excellent yields. To simplify the growth strategy for the construction of oligomers, the TBS group was deprotected to generate the final extensible chiral triazole units **4**. Three triazole building blocks bearing different side groups, including a phenyl group (**4a**), a benzyl-protected hydroxyl group (**4b**), and a tetraphenylethanyl group featuring aggregation-induced emission property (**4c**), were constructed to illustrate the feasibility of this general synthetic route, the identities of which were well confirmed by ^1^H NMR and MS analysis.

### 2.2. Construction of Stereoregular Oligotriazoles

Figure 2B illustrates the exponential growth protocol for the precise extension of the above triazole motifs to stereocontrolled oligomers, which includes three simple and mild steps (conversion of the hydroxyl group into the leaving group, deprotection of the –Boc group, and coupling of the two generated partners by Hofmann alkylation) and proved to be efficient in our previous research [29]. One different point is that, instead of transforming the hydroxyl group of **4** into halogen atoms, here it was converted into –OTs group by its reaction with 4-toluenesulfonyl chloride (TsCl), which was conducted mildly and offered the desired product **4**-OTs in a higher yield. Deprotection of the *tert*-butyl oxycarbonyl (Boc) group under acidic conditions is well-known and efficient. Hofmann alkylation of **4**-H with **4**-OTs provided dimer **5**. Through this metal-free iterative exponential growth strategy involving steps iv-vi shown in Figure 2, construction of stereoregular oligomers with exact chain-length was highly desired. To demonstrate the efficiency of this protocol, elongation of triazole units **4a** and **4b** to corresponding octamers was carried out. Related reaction details of these two growth processes and MS data of involved oligotriazoles are summarized in Table 1. All of the deprotection manipulations were terminated after a reaction time of 6 h, giving the corresponding product over a 90% yield in all cases. In parallel, sulfonylation of the hydroxyl group was finished in less than 2 h and offered the related products in yields higher than 82%. One decreasing trend of the yields in the Hofmann alkylation steps was observed with the elongation of the chain lengths. Nevertheless, the two octamers **8a** and **8b** were successfully constructed in overall yields of 7.3% and 7.6%, respectively.

The identity and high purity of all the monomers and oligomers involved in Table 1 were well verified by their own ^1^H NMR spectra provided in the Appendix A. Compounds of α-Boc-ω-OH type were also characterized by SEC, the traces of which confirmed their monodispersity as well as the elongation of the chain length after each cycle (Figure 1A,C). MS analysis of them was further conducted to illustrate the precise synthesis of these oligomers. As shown in Figure 1B,D, multiple signals of cationic adducts were observed in the mass spectra of both **8a** and **8b**, all of which were in excellent consistency with calculated molecular mass.

Encouraged by the accomplishment of two above octamers, we next explored the extension of triazole unit **4c** to stereocontrolled oligomers bearing tetraphenylethylene (TPE) groups. TPE is a typical luminophore featuring an aggregation-induced emission (AIE) phenomenon and has been widely utilized in the creation of various AIE materials [35]. Howbeit, research on the exact control of its number in precise chains is still exiguous [36,37]. As shown in Figure 3A, conversion of **4c** into **4c**-OTs and **4c**-H went smoothly, affording them in 82% and 89% yields, respectively. Subsequent coupling of **4c**-OTs and **4c**-H generated 70% of dimer **5c**, which was then split for the synthesis of **5c**-OTs and **5c**-H. Similar to the above cases, a moderate yield of 53% was observed in the fabrication of tetramer **6c** from **5c**-OTs and **5c**-H. To produce a series of oligomers bearing different amounts of TPE in an arithmetic sequence, which might be helpful in optical property studies, hexamer **7c** was created by the coupling of tetramer **6c**-H with dimer **5c**-OTs in 20% yield. The successful achievement of oligomers **5c**, **6c**, and **7c** was well confirmed by ^1^H NMR and MS characterizations, the spectra of which are all provided in the Appendix A. SEC traces shown in Figure 3B clearly demonstrated their uniformity and purity.

### 2.3. Photophysical Behaviors of TPE-Involved Oligotriazoles

UV/vis and photoluminescence (PL) characterizations were first carried out to probe the photophysical performances of these TPE-involved compounds. As shown in Figure 2A, their UV-absorption spectra exhibit three similar absorption peaks at 215, 240, and 320 nm. According to our previous study, the signal at 215 nm could be attributed to the triazole units, while the two other peaks are related to the TPE group. To reveal the impact from the oligomeric structure, solutions of **4c**–**7c** with the same concentrations of TPE groups were prepared for the investigation of their luminescence performances in solution and aggregated states. As shown in Figure 2B–E, in all cases, the photoluminescence (PL) peak intensities in THF/H_2_O mixtures with water fractions (fw) in the range of 0−80 vol % lean close to the abscissa, revealing their similar non-fluorescent feature under these conditions. This could be due to their good dispersion in these solvents, leading to the flexible stretch of TPE units and rotation of their phenyl groups. With the progressive addition of water, the AIE effect was successfully induced in all cases, in which higher PL intensity increments were observed with the elongation of chain lengths (Figure 2F). Decreased solubility in this sequence should be the main cause of this increased light emission phenomenon.

Circular dichroism (CD) and circularly polarized luminescence (CPL) analyses were further conducted to probe their chiroptical properties. Unfortunately, no obvious signals were observed in CD or CPL spectra (Figure 3A,B), which indicated their irregular folding behaviors. As shown in Figure 3C, the spatial structure of hexamer **7c** resulted from MD simulation exhibits no helical chain. This could be attributed to the large number of flexible bonds in the backbone. More strained skeletons need to be designed on the basis of this work for the development of chiroptical macromolecules with controlled circularly polarized luminescence (CPL) properties.

## 3. Material and Methods

### 3.1. General

All air- or moisture-sensitive reactions were conducted in oven-dried glassware under a nitrogen atmosphere using dry solvents. Flash column chromatography was performed over silica gel (200–300 mesh) purchased from Qingdao Puke Co., Qingdao, China. Alkynes and common organic chemicals were purchased from commercial suppliers, such as Sigma-Aldrich^®^ (Beijing, China) and J&K^®^ Scientific Ltd. (Beijing, China). and used as received. Iridium complexes were purchased from Strem^®^ Chemicals, Inc (Newburyport, USA). ^1^H NMR spectra were collected on a Bruker AV 400 MHz NMR spectrometer using residue solvent peaks as an internal standard (^1^H NMR: CDCl_3_ at 7.26 ppm, ^13^C NMR: CDCl_3_ at 77.0 ppm). HRMS (ESI) was measured on an Agilent 6540 UHD Accurate-Mass Q-TOF. SEC analyses were performed on a Waters 1525 Gel chromatography with three mixed-bed GPC columns in series (three Waters Styragel HT3 THF (7.8*300 mm Column)) and THF mobile phase run at 35 °C for 40 min. The differential refractive index of each compound was monitored using a WAT038040 (2414) detector. UV-vis absorptions were recorded using a Metash UV-6000PC. The spectra were recorded between 200 and 400 nm, with a bandwidth of 1 nm, time per point 1 s and two repetitions. **Fluorescence** performances were recorded on a Hitachi F-7000 FL Spectrophotometer. The spectra were recorded with EX slit 5.0 nm, EM slit 5.0 nm, and PMT voltage 600 V. CD analyses were performed on a Jasco-815 CD spectrometer. Solution samples were measured using a 1 mm cuvette. Sequence-defined oligomer samples (1.2 mg) were dissolved in MeOH (3 mL). CPL spectra were characterized using a JASCO CPL-200 spectrometer.

### 3.2. Preparation of 1-Thioalkynes

**1a**: At −78 °C, to a solution of phenylacetylene (50.0 mmol, 1.0 eq.) in dry THF (0.5 M) under N_2_ atmosphere was slowly added *n-*BuLi (52.0 mmol, 1.04 eq.). The reaction mixture was stirred at the same temperature for 1 h before disulfide (50.0 mmol, 1.0 eq.) was added. Then the reaction mixture was allowed warming to room temperature and stirred for 2 h before a saturated aqueous NH_4_Cl solution was added. The aqueous phase was separated and extracted with ethyl acetate (EA) three times. The combined organic phase was washed with brine, dried over Na_2_SO_4_, and evaporated under vacuum to give the crude product, which was then purified by silica gel flash column chromatography to give 91% of pure thioalkyne **1a** as colorless oil (45.5 mmol, 13.3 g). Rf = 0.5 (PE/EA = 50/1). ^1^H NMR (400 MHz, CDCl_3_) δ 7.44–7.41 (m, 2 H), 7.34–7.31 (m, 3 H), 3.99 (t, *J* = 12.0 Hz, 2 H), 2.95 (t, *J* = 12.0 Hz, 2 H), 0.95 (s, 9 H), 0.14 (s, 6 H). ^13^C NMR (100 MHz, CDCl_3_) δ 131.4, 128.3, 128.0, 123.4, 92.6, 79.2, 61.7, 38.0, 25.9, 18.4, −5.3.

**1b**: (i) At −78 °C, to a solution of propargyl alcohol (60.0 mmol, 1.0 eq.) in dry THF (0.25 M) under N_2_ atmosphere was slowly added *n-*BuLi (126 mmol, 2.1 eq.). The reaction mixture was stirred at the same temperature for 1 h before disulfide (1.0 eq.) was added. Then the reaction mixture was allowed warming to room temperature and stirred for 2 h before a saturated aqueous NH_4_Cl solution was added. The aqueous phase was separated and extracted with ethyl acetate (EA) three times. The combined organic phase was washed with brine, dried over Na_2_SO_4_, and evaporated under vacuum to give the crude product, which was then purified by silica gel flash column chromatography to give 88% of pure thioalkyne **1**-OH as colorless oil (52.8 mmol). (ii) At 0 °C, to a solution of the obtained thioalkyne (25.0 mmol, 1.0 eq.) in dry THF (0.5 M) under N_2_ atmosphere was slowly added NaH (30.0 mmol, 1.2 eq.). The reaction mixture was stirred at room temperature for 4 h until the reaction completed, which was confirmed by TLC. The reaction mixture was cooled at 0 °C and water was added. The aqueous phase was separated and extracted with EA three times. The combined organic phase was washed with brine, dried over Na_2_SO_4_, and evaporated under vacuum to give the crude product, which was then purified by silica gel flash column chromatography to give 87% of pure thioalkyne **1b** as colorless oil (21.7 mmol, 7.3 g). Rf = 0.4 (PE/EA = 50/1). ^1^H NMR (400 MHz, CDCl_3_) δ 7.39–7.30 (m, 5 H), 4.59 (s, 2 H), 4.27 (s, 2 H), 3.90 (t, *J* = 12.0 Hz, 2 H), 2.84 (t, *J* = 12.0 Hz, 2 H), 0.91 (s, 9 H), 0.09 (s, 6 H). ^13^C NMR (100 MHz, CDCl_3_) δ 137.4, 128.4, 128.1, 127.8, 89.8, 77.1, 71.3, 61.7, 58.2, 37.8, 25.8, 18.3, −5.3.

**1c**: Thioalkyne **1**-OH (25.0 mmol, 1.0 eq.) was dissolved in dry DCM (0.5 M), and PPh_3_ (27.5 mmol, 1.1 eq.), imidazole (27.5 mmol, 1.1 eq.), and iodine (27.5 mmol, 1.1 eq.) were added at 0 °C and stirred for 40 min. Upon completion indicated by TLC, aqueous Na_2_S_2_O_3_ solution was added and extracted with EA. The organic layer was dried over MgSO_4_ and evaporated under vacuum to give the crude iodination product, which was then dissolved in MeCN with K_2_CO_3_ (37.5 mmol, 1.5 eq.) and TPE-OH (25.0 mmol, 1.0 eq.) and stirred at 60 °C till completion. Removal of solvent and purification with column chromatography afforded 66% of pure thioalkyne **1c** as colorless oil (16.5 mmol, 9.5 g). Rf = 0.4 (PE/EA = 10/1). ^1^H NMR (400 MHz, CDCl_3_) δ 7.10–7.00 (m, 15 H), 6.95–6.92 (m, 2 H), 6.68–6.66 (m, 2 H), 4.69 (s, 2 H), 3.86 (t, *J* = 12.0 Hz, 2 H), 2.81 (t, *J* = 12.0 Hz, 2 H), 0.90 (s, 9 H), 0.07 (s, 6 H). ^13^C NMR (100 MHz, CDCl_3_) δ 156.2, 143.9, 143.8, 140.3, 140.3, 136.7, 132.5, 131.4, 131.3, 128.5, 128.3, 128.1, 127.7, 127.6, 126.4, 126.3, 126.2, 125.8, 114.0, 88.7, 78.4, 61.5, 56.7, 37.8, 25.8, 18.3, −5.3.

### 3.3. Synthesis of Triazole Units

General procedure for IrAAC: in a glove box, to an oven-dried vial was added thioalkyne **1** (1.0 eq.), chiral azide **2** (1.2 eq.), [Ir(COD)Cl]_2_ (2 mol %), and THF (0.5 M). The vial was capped and removed from the glove box. The reaction mixture was stirred at room temperature for 2–8 h until the reaction was completed (confirmed by TLC) and then concentrated under reduced pressure. The residue was purified by silica gel flash column chromatography to give the desired product.

**3a**: obtained from the reaction of **1a** (40.0 mmol) with **2** (48.0 mmol) in 92% yield (19.1 g, 36.8 mmol) as colorless oil.

Rf = 0.5 (PE/EA = 3/1).

^1^H NMR (400 MHz, CDCl_3_) δ 8.13–8.10 (m, 2 H), 7.38–7.26 (m, 3 H), 4.61–4.41 (m, 2 H), 4.31–4.26 (m, 1 H), 3.55–3.19 (m, 4 H), 2.72–2.66 (m, 2 H), 1.85–1.65 (m, 4 H), 1.40–1.37 (m, 9 H), 0.76 (s, 9 H), 0.12 (s, 6 H).

^13^C NMR (100 MHz, CDCl_3_) δ 154.2, 147.8, 130.5, 128.4, 128.2, 126.6, 125.9, 79.9, 79.3, 61.5, 56.8, 56.4, 49.9, 49.5, 46.6, 46.2, 38.2, 28.3, 27.6, 25.6, 23.2, 22.4, 18.0, −5.5, −5.6.

**3b**: obtained from the reaction of **1b** (20.0 mmol) with **2** (24.0 mmol) in 92% yield (10.9 g, 19.4 mmol) as colorless oil. 

Rf = 0.5 (PE/EA = 3/1).

^1^H NMR (400 MHz, CDCl_3_) δ 8.13–8.10 (m, 2 H), 7.38–7.26 (m, 3 H), 4.61–4.41 (m, 2 H), 4.31–4.26 (m, 1 H), 3.55–3.19 (m, 4 H), 2.72–2.66 (m, 2 H), 1.85–1.65 (m, 4 H), 1.40–1.37 (m, 9 H), 0.76 (s, 9 H), 0.12 (s, 6 H).

^13^C NMR (100 MHz, CDCl_3_) δ 147.5, 137.8, 130.0, 128.2, 127.8, 127.5, 80.0, 79.4, 72.2, 62.7, 61.8, 56.4, 49.8, 46.6, 46.2, 39.5, 28.4, 27.7, 25.8, 25.7, 23.3, 22.4, 18.1, −5.4, −5.5.

**3c**: obtained from the reaction of **1c** (15.0 mmol) with **2** (18.0 mmol) in 79% yield (9.52 g, 11.85 mmol) as colorless oil.

Rf = 0.4 (PE/EA = 2/1).

^1^H NMR (400 MHz, CDCl_3_) δ 7.15–7.02 (m, 15 H), 6.97–6.95 (m, 2 H), 6.79–6.76 (m, 2 H), 5.12 (s, 2 H), 4.64–4.32 (m, 3 H), 3.70–3.67 (m, 2 H), 3.44–3.27 (m, 2 H), 2.93–2.89 (m, 2 H), 1.89–1.76 (m, 4 H), 1.48 (s, 9 H), 0.87 (s, 9 H), 0.01 (s, 6 H).

^13^C NMR (100 MHz, CDCl_3_) δ 143.9, 140.2, 132.5, 131.3, 130.5, 127.7, 127.6, 126.3, 113.9, 61.8, 61.0, 56.8, 28.5, 25.8, 25.6, 18.2, −3.6, −5.4.

General procedure for deprotection of TBS group: The TBS-protected triazole product was dissolved in THF (0.5 M). TBAF.3H_2_O (1.05 eq.) was added at 0 °C and stirred till completion indicated by TLC. EA was then added and the mixture was washed by water and brine. The organic layer was dried over MgSO_4_ and evaporated under vacuum to give the crude product, which was then purified by silica gel flash column chromatography to give pure product.

**4a**: obtained from the reaction of **3a** (35.0 mmol) in 96% yield (13.6 g, 33.6 mmol) as colorless oil.

Rf = 0.3 (PE/EA = 2/1).

^1^H NMR (400 MHz, CDCl_3_) δ 8.14–8.12 (m, 2 H), 7.45–7.41 (m, 2 H), 7.37–7.33 (m, 1 H), 4.83–4.79 (m, 1 H), 4.56–4.22 (m, 2 H), 3.58–3.46 (m, 2 H), 3.37–3.26 (m, 2 H), 2.88–2.76 (m, 2 H), 2.14–2.12 (m, 1 H), 1.93–1.65 (m, 4 H), 1.46–1.40 (m, 9 H).

^13^C NMR (100 MHz, CDCl_3_) δ 154.8, 154.3, 148.3, 130.3, 128.4, 128.3, 126.6, 125.9, 80.0, 60.0, 56.8, 56.6, 50.0, 49.4, 46.6, 46.1, 39.5, 38.1, 28.2, 27.8, 23.0, 22.3.

HRMS *m/z* (ESI) calcd. for C_20_H_29_N_4_O_3_S (M+H)^+^ 405.1995, found 405.1999.

**4b**: obtained from the reaction of **3b** (18.0 mmol) in 92% yield (7.44 g, 16.56 mmol) as colorless oil.

Rf = 0.3 (PE/EA = 2/1).

^1^H NMR (400 MHz, CDCl_3_) δ 7.29–7.21 (m, 5 H), 4.68–4.56 (m, 5 H), 4.40–4.17 (m, 2 H), 3.53–3.51 (m, 2 H), 3.26–3.24 (m, 2 H), 2.90–2.83 (m, 2 H), 1.96–1.77 (m, 4 H), 1.41 (s, 9 H).

^13^C NMR (100 MHz, CDCl_3_) δ 154.5, 154.0, 147.8, 147.5, 137.2, 129.2, 128.1, 127.7, 127.5, 79.7, 72.1, 62.4, 59.9, 56.4, 49.9, 49.5, 46.5, 46.0, 39.9, 38.8, 28.1, 27.7, 22.9, 22.2.

HRMS *m/z* (ESI) calcd. for C_22_H_33_N_4_O_4_S (M+H)^+^ 449.2223, found 449.2226.

**4c**: obtained from the reaction of **3c** (10.0 mmol) in 90% yield (6.20 g, 9.0 mmol) as colorless oil.

Rf = 0.2 (PE/EA = 2/1).

^1^H NMR (400 MHz, CDCl_3_) δ 7.12–7.00 (m, 15 H), 6.96–6.93 (m, 2 H), 6.77–6.74 (m, 2 H), 5.11 (s, 2 H), 4.81–4.77 (m, 1 H), 4.44–4.40 (m, 1 H), 4.28–4.24 (m, 1 H), 3.65–3.62 (m, 2 H), 3.34–3.28 (m, 2 H), 2.99–2.91 (m, 2 H), 2.10–2.08 (m, 1 H), 1.91–1.83 (m, 3 H), 1.68–1.63 (m, 1 H), 1.47–1.46 (m, 9 H).

^13^C NMR (100 MHz, CDCl_3_) δ 156.7, 155.0, 147.2, 143.8, 143.8, 140.3, 136.7, 132.5, 131.2, 127.7, 127.5, 126.3, 126.2, 113.9, 85.5, 80.3, 61.1, 60.3, 56.7, 49.8, 46.8, 40.9, 28.42 28.0, 23.2.

HRMS *m/z* (ESI) calcd. for C_41_H_44_N_4_O_4_SNa (M+Na)^+^ 711.2975, found 711.2982.

### 3.4. Precise Construction of Oligotriazoles

General procedure for sulfonylation: Compound **X** was dissolved in DCM (0.5 M) with subsequent addition of Et_3_N (2.0 eq.) and 4-dimethylaminopyridine (DMAP, 1 mol %). Then the solution of 4-toluenesulfonyl chloride (TsCl, 1.5 eq.) in DCM (1.0 M) was slowly added into the previous mixture. The reaction mixture was stirred at room temperature until completion confirmed by TLC, and then washed with brine (three times), dried over Na_2_SO_4_, filtered, and evaporated under vacuum to give the residue, which was then purified by silica gel flash column chromatography to give pure product **X**-OTs.

General procedure for deprotection of –Boc group: At 0 °C, compound **X** was dissolved in methanol (0.5 M), with the subsequent slow addition of the solution of acetyl chloride (AcCl, 3.0 eq.) in MeOH (1.5 M). The reaction mixture was stirred at room temperature until completion confirmed by TLC, and then evaporated under vacuum. The residue was diluted with saturated Na_2_CO_3_ aqueous solution, washed with DCM (three times). The organic layer was dried over Na_2_SO_4_ and evaporated under vacuum to give the residue, which was then purified by silica gel flash column chromatography to give pure product **X**-H.

General procedure for Hofmann alkylation: **X**-H (1.0 eq.) was dissolved in acetonitrile (0.5 M), with the subsequent addition of K_2_CO_3_ (1.5 eq.). The reaction mixture was stirred at room temperature for 30 min before the addition of **X**-OTs (1.1 eq.). The reaction mixture was stirred at 80 °C until the reaction completed, which was confirmed by TLC. The solution was cooled to room temperature and filtered. The residue was evaporated under vacuum and purified by column chromatography on silica gel to give pure product.

**4a**-OTs: obtained from the reaction of **4a** (15.0 mmol) in 87% yield (7.26 g, 13.0 mmol) as colorless oil.

Rf = 0.6 (DCM/MeOH = 30/1).

^1^H NMR (400 MHz, CDCl_3_) δ 8.10–8.08 (m, 2 H), 7.63–7.61 (m, 2 H), 7.47–7.38 (m, 3 H), 7.31–7.28 (m, 2 H), 4.60–4.51 (m, 2 H), 4.33–4.29 (m, 1 H), 3.94–3.84 (m, 2 H), 3.38–3.27 (m, 2 H), 2.91–2.81 (m, 2 H), 2.44 (s, 3 H), 1.94–1.65 (m, 4 H), 1.49–1.43 (m, 9 H).

^13^C NMR (100 MHz, CDCl_3_) δ 132.4, 129.9, 128.6, 127.7, 126.8, 126.6, 124.5, 79.6, 67.4, 67.1, 57.0, 56.5, 50.3, 49.8, 46.3, 34.1, 28.3, 21.5.

**4a**-H: obtained from the reaction of **4a** (15.0 mmol) in 96% yield (4.38 g, 14.4 mmol) as colorless oil.

Rf = 0.2 (DCM/MeOH = 20/1).

^1^H NMR (400 MHz, CDCl_3_) δ 8.16–8.14 (m, 2 H), 7.46–7.35 (m, 3 H), 4.66–4.60 (m, 1 H), 4.24–4.20 (m, 1 H), 4.14–4.08 (m, 1 H), 3.50–3.44 (m, 1 H), 3.09–3.03 (m, 1 H), 2.95–2.83 (m, 3 H), 2.71–2.64 (m, 1 H), 2.14–2.11 (m, 1 H), 1.92–1.75 (m, 2 H), 1.64–1.57 (m, 1 H).

^13^C NMR (100 MHz, CDCl_3_) δ 148.0, 130.6, 128.7, 128.4, 126.6, 125.3, 57.7, 57.3, 52.5, 46.3, 37.5, 29.4, 25.5.

**5a**: obtained from the reaction of **4a**-OTs (11.0 mmol) with **4a**-H (10.0 mmol) in 68% yield (4.7 g, 6.8 mmol) as colorless oil.

Rf = 0.4 (DCM/MeOH = 30/1).

^1^H NMR (400 MHz, CDCl_3_) δ 8.15–8.11 (m, 4 H), 7.42–7.29 (m, 6 H), 4.68–4.12 (m, 4 H), 3.90–3.62 (m, 1 H), 3.47–3.20 (m, 4 H), 2.89–2.66 (m, 7 H), 2.41–2.39 (m, 1 H), 2.07–2.05 (m, 1 H), 1.87–1.63 (m, 8 H), 1.43–1.36 (m, 9 H).

^13^C NMR (100 MHz, CDCl_3_) δ 154.7, 148.0, 130.5, 128.5, 128.3, 126.6, 126.5, 1263, 125.3, 80.0, 79.7, 63.5, 59.7, 57.0, 56.5, 54.0, 53.5, 51.6, 50.2, 49.6, 46.7, 46.2, 38.6, 34.2, 33.95, 28.5, 28.3, 27.5, 23.2, 22.4.

HRMS *m/z* (ESI) calcd. for C_35_H_46_N_8_O_3_S_2_Na (M+Na)^+^ 713.3026, found 713.3039.

**5a**-OTs: obtained from the reaction of **5a** (3.0 mmol) in 86% yield (2.18 g, 2.58 mmol) as colorless oil.

Rf = 0.5 (DCM/MeOH = 30/1).

^1^H NMR (400 MHz, CDCl_3_) δ 8.19–8.17 (m, 2 H), 8.08–8.06 (m, 2 H), 7.61–7.59 (m, 2 H), 7.46–7.35 (m, 6 H), 7.29–7.27 (m, 2 H), 4.70–4.32 (m, 3 H), 4.29–4.20 (m, 2 H), 3.82 (t, *J* = 12.0 Hz, 2 H), 3.42–3.27 (m, 2 H), 2.94–2.89 (m, 2 H), 2.78–2.66 (m, 5 H), 2.43 (s, 3 H), 2.12–2.10 (m, 1 H), 1.89–1.65 (m, 9 H), 1.46–1.42 (m, 9 H).

^13^C NMR (100 MHz, CDCl_3_) 154.5, 147.9, 145.0, 132.4, 130.7, 130.3, 129.9, 128.6, 128.5, 128.3, 127.7, 126.7, 126.6, 126.2, 124.7, 80.0, 79.5, 77.2, 67.3, 63.6, 57.0, 56.5, 54.4, 53.6, 51.6, 50.1, 49.6, 46.7, 46.3, 34.4, 34.1, 28.5, 28.4, 23.4, 22.5, 21.6.

**5a**-H: obtained from the reaction of **5a** (3.0 mmol) in 96% yield (1.70 g, 2.88 mmol) as colorless oil.

Rf = 0.5 (DCM/MeOH = 15/1).

^1^H NMR (400 MHz, CDCl_3_) δ 8.15–8.12 (m, 4 H), 7.43–7.31 (m, 6 H), 4.42–4.35 (m, 3 H), 4.22–4.17 (m, 1 H), 3.65–3.62 (m, 1 H), 3.44–3.41 (m, 2 H), 2.95–2.78 (m, 10 H), 2.47–2.42 (m, 4 H), 2.11–2.06 (m, 1 H), 1.86–1.49 (m, 9 H).

^13^C NMR (100 MHz, CDCl_3_) 148.3, 130.6, 128.7, 128.5, 126.9, 126.7, 126.0, 63.7, 59.5, 57.8, 54.6, 53.8, 52.6, 51.7, 46.3, 38.9, 34.5, 29.2, 28.7, 25.0, 23.4.

**6a**: obtained from the reaction of **5a**-OTs (2.2 mmol) with **5a**-H (2.0 mmol) in 52% yield (1.315 g, 1.04 mmol) as colorless oil.

Rf = 0.4 (DCM/MeOH = 20/1).

^1^H NMR (400 MHz, CDCl_3_) δ 8.16–8.11 (m, 8 H), 7.43–7.31 (m, 12 H), 4.67–4.11 (m, 8 H), 3.73–3.70 (m, 1 H), 3.43–3.25 (m, 4 H), 2.90–2.65 (m, 18 H), 2.38–2.33 (m, 2 H), 2.10–2.03 (m, 4 H), 1.68–1.60 (m, 16 H), 1.44–1.39 (m, 9 H).

HRMS *m/z* (ESI) calcd. for C_65_H_84_N_16_O_3_S_4_ (M+2H)^2+^ 632.2892, found 632.2893.

**6a**-OTs: obtained from the reaction of **6a** (0.4 mmol) in 82% yield (465.1 mg, 0.328 mmol) as colorless oil.

Rf = 0.5 (DCM/MeOH = 20/1).

^1^H NMR (400 MHz, CDCl_3_) δ 8.16–8.12 (m, 6 H), 8.04–8.02 (m, 2 H), 7.57–7.55 (m, 2 H), 7.43–7.33 (m, 12 H), 7.26–7.24 (m, 2 H), 4.67–4.49 (m, 9 H), 3.77 (t, *J* = 12.0 Hz, 2 H), 3.33–3.26 (m, 2 H), 2.94–2.60 (m, 16 H), 2.40–2.34 (m, 6 H), 2.10–1.58 (m, 20 H), 1.44–1.40 (m, 9 H).

**6a**-H: obtained from the reaction of **6a** (0.4 mmol) in 90% yield (419.0 mg, 0.36 mmol) as colorless oil.

Rf = 0.2 (DCM/MeOH = 15/1).

^1^H NMR (400 MHz, CDCl_3_) δ 8.15–8.10 (m, 8 H), 7.44–7.33 (m, 12 H), 4.45–4.38 (m, 2 H), 4.30–4.12 (m, 6 H), 3.72–3.66 (m, 2 H), 3.41–3.36 (m, 2 H), 3.01–2.83 (m, 10 H), 2.76–2.61 (m, 10 H), 2.40–2.34 (m, 3 H), 2.12–2.03 (m, 3 H), 1.93–1.50 (m, 16 H).

**8a**: obtained from the reaction of **6a**-OTs (0.22 mmol) with **6a**-H (0.2 mmol) in 25% yield (120.5 mg, 0.05 mmol) as colorless oil.

Rf = 0.2 (DCM/MeOH = 20/1).

^1^H NMR (400 MHz, CDCl_3_) δ 8.15–8.11 (m, 16 H), 7.34–7.32 (m, 24 H), 4.70–4.57 (m, 2 H), 4.40–4.11 (m, 16 H), 3.42–3.24 (m, 4 H), 2.91–2.58 (m, 32 H), 2.34–2.31 (m, 6 H), 2.08–2.04 (m, 8 H), 1.86–1.61 (m, 36 H), 1.44–1.40 (m, 9 H).

HRMS *m/z* (ESI) calcd. for C_125_H_156_N_32_O_3_S_8_ (M+2H)^2+^ 1205.0420, found 1205.0420.

**4b**-OTs: obtained from the reaction of **4b** (7.5 mmol) in 84% yield (3.8 g, 6.3 mmol) as colorless oil.

Rf = 0.6 (DCM/MeOH = 30/1).

^1^H NMR (400 MHz, CDCl_3_) δ 7.69–7.66 (m, 2 H), 7.32–7.26 (m, 7 H), 4.62–4.50 (m, 6 H), 4.25–3.96 (m, 3 H), 3.37–3.24 (m, 2 H), 3.02–3.00 (m, 2 H), 2.42 (s, 3 H), 3.31–3.25 (m, 1 H), 1.88–1.78 (m, 4 H), 1.42 (s, 9 H).

^13^C NMR (100 MHz, CDCl_3_) δ 147.8, 145.0, 137.5, 132.5, 129.8, 128.3, 127.8, 127.7, 80.0, 79.4, 72.1, 67.7, 67.4, 62.6, 56.4, 50.4, 46.5, 35.2, 34.9, 28.3, 23.2, 22.4, 21.5.

**4b**-H: obtained from the reaction of **4b** (7.5 mmol) in 92% yield (2.4 g, 6.9 mmol) as colorless oil.

Rf = 0.2 (DCM/MeOH = 20/1).

^1^H NMR (400 MHz, CDCl_3_) δ 7.35–7.26 (m, 5 H), 4.68–4.66 (m, 2 H), 4.59–4.51 (m, 3 H), 4.27–4.19 (m, 3 H), 4.00–3.97 (m, 1 H), 3.57–3.52 (m, 1 H), 3.31–3.25 (m, 1 H), 3.06–3.01 (m, 1 H), 2.94 (t, *J* = 12.0 Hz, 2 H), 2.83–2.77 (m, 1 H), 2.08–2.03 (m, 1 H), 1.89–1.75 (m, 1 H), 1.59–1.54 (m, 1 H).

^13^C NMR (100 MHz, CDCl_3_) δ 147.5, 137.4, 129.1, 128.2, 127.7, 127.5, 72.1, 62.7, 58.5, 57.3, 52.2, 46.1, 39.0, 29.1, 25.1.

**5b**: obtained from the reaction of **4b**-OTs (5.5 mmol) with **4b**-H (5.0 mmol) in 76% yield (2.96 g, 3.8 mmol) as colorless oil.

Rf = 0.4 (DCM/MeOH = 30/1).

^1^H NMR (400 MHz, CDCl_3_) δ 7.37–7.28 (m, 10 H), 4.69–4.61 (m, 10 H), 4.29–4.14 (m, 4 H), 3.57–3.49 (m, 6 H), 3.05–2.85 (m, 5 H), 2.50–2.49 (m, 1 H), 2.15–1.68 (m, 9 H), 1.44 (s, 9 H).

^13^C NMR (100 MHz, CDCl_3_) δ 154.6, 147.6, 137.6, 130.0, 128.9, 128.3, 128.2, 127.9, 127.9, 127.7, 80.0, 79.7, 72.3, 63.4, 62.7, 62.6, 59.8, 56.4, 54.2, 53.5, 51.8, 49.9, 46.6, 39.5, 39.2, 35.1, 28.5, 28.3, 27.6, 25.6, 23.1, 22.4, 17.9, −3.7.

HRMS *m/z* (ESI) calcd. for C_39_H_55_N_8_O_5_S_2_ (M+H)^+^ 801.3551, found 801.3547.

**5b**-OTs: obtained from the reaction of **5b** (1.5 mmol) in 82% yield (1.15 g, 1.23 mmol) as colorless oil.

Rf = 0.5 (DCM/MeOH = 30/1).

^1^H NMR (400 MHz, CDCl_3_) δ 7.68–7.66 (m, 2 H), 7.36–7.25 (m, 12 H), 4.68–4.53 (m, 10 H), 4.42–4.40 (m, 1 H), 4.31–4.28 (m, 1 H), 4.20–3.93 (m, 4 H), 3.36–3.24 (m, 3 H), 3.04–2.79 (m, 6 H), 2.41 (s, 3 H), 2.14–2.08 (m, 1 H), 1.89–1.64 (m, 8 H), 1.44 (s, 9 H).

^13^C NMR (100 MHz, CDCl_3_) δ 147.5, 145.0, 137.5, 132.5, 129.8, 128.5, 128.3, 128.2, 127.9, 127.8, 127.7, 72.1, 67.6, 63.4, 62.8, 62.6, 54.4, 53.4, 51.7, 35.2, 28.3, 23.3, 21.5.

**5b**-H: obtained from the reaction of **5b** (1.5 mmol) in 94% yield (957.4 mg, 1.41 mmol) as colorless oil.

Rf = 0.5 (DCM/MeOH = 15/1).

^1^H NMR (400 MHz, CDCl_3_) δ 7.36–7.27 (m, 10 H), 4.69–4.60 (m, 8 H), 4.40–4.30 (m, 3 H), 4.18–4.13 (m, 2 H), 3.67–3.47 (m, 4 H), 2.96–2.81 (m, 9 H), 2.53–2.48 (m, 1 H), 2.14–2.12 (m, 1 H), 1.90–1.50 (m, 10 H).

^13^C NMR (100 MHz, CDCl_3_) δ 147.7, 137.6, 137.4, 129.6, 129.1, 128.3, 128.2, 127.9, 127.8, 127.7, 127.6, 72.4, 72.3, 63.7, 63.4, 62.8, 62.7, 59.6, 57.7, 54.6, 53.6, 52.7, 51.9, 46.1, 39.6, 35.3, 29.5, 29.0, 28.6, 24.9, 23.2.

**6b**: obtained from the reaction of **5b**-OTs (1.1 mmol) with **5b**-H (1.0 mmol) in 59% yield (849.6 mg, 0.59 mmol) as colorless oil.

Rf = 0.4 (DCM/MeOH = 20/1).

^1^H NMR (400 MHz, CDCl_3_) δ 7.36–7.25 (m, 20 H), 4.68–4.61 (m, 18 H), 4.40–4.08 (m, 8 H), 3.35–3.24 (m, 6 H), 2.93–2.79 (m, 16 H), 2.44–2.42 (m, 2 H), 2.13–2.11 (m, 2 H), 1.85–1.65 (m, 16 H), 1.44 (s, 9 H).

HRMS *m/z* (ESI) calcd. for C_73_H_100_N_16_O_7_S_4_ (M+2H)^2+^ 720.3416, found 720.3419.

**6b**-OTs: obtained from the reaction of **6b** (0.2 mmol) in 86% yield (274.2 mg, 0.172 mmol) as colorless oil.

Rf = 0.5 (DCM/MeOH = 20/1).

^1^H NMR (400 MHz, CDCl_3_) δ 7.66–7.64 (m, 2 H), 7.36–7.24 (m, 22 H), 4.68–4.51 (m, 18 H), 4.35–4.15 (m, 11 H), 3.35–3.25 (m, 3 H), 2.96–2.73 (m, 15 H), 2.47–2.40 (m, 6 H), 2.13–2.10 (m, 2 H), 1.86–1.59 (m, 16 H), 1.44 (s, 9 H).

**6b**-H: obtained from the reaction of **6b** (0.2 mmol) in 90% yield (241.2 mg, 0.18 mmol) as colorless oil.

Rf = 0.2 (DCM/MeOH = 15/1).

^1^H NMR (400 MHz, CDCl_3_) δ 7.35–7.26 (m, 20 H), 4.65–4.60 (m, 18 H), 4.31–4.07 (m, 6 H), 3.87–3.83 (m, 1 H), 3.48 (t, *J* = 12.0 Hz, 2 H), 3.17–3.05 (m, 4 H), 2.98–2.77 (m, 15 H), 2.48–2.41 (m, 2 H), 2.14–2.09 (m, 2 H), 1.90–1.59 (m, 17 H), 1.36 (t, *J* = 16.0 Hz, 3 H).

**8b**: obtained from the reaction of **6b**-OTs (0.11 mmol) with **6b**-H (0.1 mmol) in 22% yield (60.8 mg, 0.022 mmol) as colorless oil.

Rf = 0.2 (DCM/MeOH = 20/1).

^1^H NMR (400 MHz, CDCl_3_) δ 7.37–7.27 (m, 40 H), 4.69–4.55 (m, 36 H), 4.24–4.11 (m, 16 H), 3.49–3.27 (m, 8 H), 2.95–2.77 (m, 32 H), 2.45–2.43 (m, 6 H), 2.14–2.11 (m, 6 H), 1.86–1.62 (m, 32 H), 1.45 (s, 9 H).

HRMS *m/z* (ESI) calcd. for C_141_H_186_N_32_O_11_S_8_Na_2_ (M+2Na)^2+^ 1403.6387, found 1403.6356.

**4c**-OTs: obtained from the reaction of **4c** (4.0 mmol) in 82% yield (2.77 g, 3.28 mmol) as colorless oil.

Rf = 0.5 (DCM/MeOH = 30/1).

^1^H NMR (400 MHz, CDCl_3_) δ 7.69–7.67 (m, 2 H), 7.31–7.28 (m, 2 H), 7.16–7.06 (m, 15 H), 6.97–6.95 (m, 2 H), 6.70–6.68 (m, 2 H), 5.05 (s, 2 H), 4.58–4.47 (m, 2 H), 4.30–4.26 (m, 1 H), 4.10–4.04 (m, 2 H), 3.44–3.25 (m, 2 H), 3.05–3.02 (m, 2 H), 2.42 (s, 3 H), 1.92–1.81 (m, 4 H), 1.46 (s, 9 H).

^13^C NMR (100 MHz, CDCl_3_) δ 143.9, 132.5, 131.3, 130.0, 127.8, 127.7, 127.6, 126.3, 113.8, 28.4, 21.6.

**4c**-H: obtained from the reaction of **4c** (4.0 mmol) in 89% yield (2.21 g, 3.76 mmol) as colorless oil.

Rf = 0.2 (DCM/MeOH = 20/1).

^1^H NMR (400 MHz, CDCl_3_) δ 7.12–7.00 (m, 15 H), 6.95–6.93 (m, 2 H), 6.74–6.72 (m, 2 H), 5.16–5.07 (m, 2 H), 4.61–4.55 (m, 1 H), 4.23–4.19 (m, 1 H), 4.09–4.07 (m, 1 H), 3.58–3.55 (m, 1 H), 3.17–3.11 (m, 2 H), 2.96–2.92 (m, 2 H), 2.82–2.77 (m, 1 H), 2.13–2.10 (m, 1 H), 1.91–1.78 (m, 2 H), 1.62–1.57 (m, 1 H).

^13^C NMR (100 MHz, CDCl_3_) δ 156.7, 146.9, 143.8, 140.3, 136.8, 132.6, 131.3, 129.8, 127.7, 127.6, 126.3, 126.2, 113.8, 61.2, 58.4, 57.5, 51.9, 46.2, 39.3, 29.2, 25.1.

**5c**: obtained from the reaction of **4c**-OTs (2.75 mmol) with **4c**-H (2.5 mmol) in 70% yield (2.2 g, 1.75 mmol) as colorless oil.

Rf = 0.3 (DCM/MeOH = 30/1).

^1^H NMR (400 MHz, CDCl_3_) δ 7.11–7.02 (m, 30 H), 6.97–6.95 (m, 4 H), 6.78–6.75 (m, 4 H), 5.12–5.10 (m, 4 H), 4.70–4.43 (m, 2 H), 4.31–4.20 (m, 2 H), 3.55–3.35 (m, 2 H), 3.22–3.19 (m, 3 H), 2.95–2.90 (m, 7 H), 2.55–2.53 (m, 1 H), 2.18–2.16 (m, 2 H), 1.77–1.68 (m, 8 H), 1.47 (s, 9 H).

^13^C NMR (100 MHz, CDCl_3_) δ 156.8, 143.8, 140.3, 132.5, 131.3, 128.4, 127.7, 127.6, 126.3, 126.2, 113.8, 61.2, 28.7, 28.5, 23.3.

HRMS *m/z* (ESI) calcd. for C_77_H_79_N_8_O_5_S_2_ (M+H)^+^ 1259.5609, found 1259.5607.

**5c**-OTs: obtained from the reaction of **5c** (0.7 mmol) in 79% yield (777.8 mg, 0.553 mmol) as colorless oil.

Rf = 0.4 (DCM/MeOH = 30/1).

^1^H NMR (400 MHz, CDCl_3_) δ 7.67–7.65 (m, 2 H), 7.31–7.03 (m, 32 H), 6.97–6.94 (m, 4 H), 6.79–6.76 (m, 2 H), 6.68–6.66 (m, 2 H), 5.11 (s, 2 H), 5.02 (s, 2 H), 4.68–4.57 (m, 2 H), 4.34–4.25 (m, 4 H), 4.02–3.98 (m, 2 H), 3.43–3.25 (m, 2 H), 2.99–2.81 (m, 8 H), 2.53–2.41 (m, 4 H), 1.91–1.68 (m, 7 H), 1.46 (s, 9 H).

^13^C NMR (100 MHz, CDCl_3_) δ 143.9, 132.5, 131.3, 130.0, 127.8, 127.7, 127.6, 126.3, 113.8, 61.0, 28.5, 21.6.

**5c**-H: obtained from the reaction of **5c** (0.7 mmol) in 87% yield (707.6 mg, 0.61 mmol) as colorless oil.

Rf = 0.4 (DCM/MeOH = 15/1).

H NMR (400 MHz, CDCl_3_) δ 7.13–7.00 (m, 30 H), 6.97–6.94 (m, 4 H), 6.77–6.73 (m, 4 H), 5.10–5.08 (m, 4 H), 4.51–4.49 (m, 2 H), 4.43–4.38 (m, 1 H), 4.26–4.22 (m, 1 H), 3.77–3.73 (m, 1 H), 3.54–3.51 (m, 2 H), 3.18–3.16 (m, 2 H), 3.03–2.83 (m, 8 H), 2.56–2.53 (m, 1 H), 2.18–2.16 (m, 1 H), 1.91–1.58 (m, 9 H).

^13^C NMR (100 MHz, CDCl_3_) δ 143.9, 140.3, 132.6, 131.3, 127.7, 127.6, 126.3, 113.9, 46.2, 29.7, 29.1.

**6c**: obtained from the reaction of **5c**-OTs (0.33 mmol) with **5c**-H (0.30 mmol) in 53% yield (384.2 mg, 0.159 mmol) as colorless oil.

Rf = 0.3 (DCM/MeOH = 20/1).

^1^H NMR (400 MHz, CDCl_3_) δ 7.17–7.01 (m, 60 H), 6.97–6.94 (m, 8 H), 6.77–6.74 (m, 8 H), 5.11–5.06 (m, 8 H), 4.74–4.45 (m, 2 H), 4.31–4.17 (m, 8 H), 3.54–3.22 (m, 6 H), 2.96–2.77 (m, 16 H), 2.50–2.44 (m, 2 H), 2.17–2.14 (m, 2 H), 1.87–1.63 (m, 16 H), 1.44 (s, 9 H).

HRMS *m/z* (ESI) calcd. for C_149_H_148_N_16_O_7_S_4_ (M+2H)^2+^ 1201.5373, found 1201.5366.

**6c**-H: obtained from the reaction of **6c** (0.1 mmol) in 88% yield (202.4 mg, 0.088 mmol) as colorless oil.

Rf = 0.3 (DCM/MeOH = 20/1).

^1^H NMR (400 MHz, CDCl_3_) δ 7.14–6.95 (m, 68 H), 6.77–6.65 (m, 8 H), 5.10–5.05 (m, 8 H), 4.53–4.17 (m, 8 H), 3.85–3.72 (m, 1 H), 3.60–3.48 (m, 2 H), 3.05–2.73 (m, 16 H), 2.54–2.11 (m, 12 H), 1.85–1.56 (m, 15 H).

**7c**: obtained from the reaction of **5c**-OTs (0.055 mmol) with **6c**-H (0.05 mmol) in 20% yield (35.4 mg, 0.01 mmol) as colorless oil.

Rf = 0.2 (DCM/MeOH = 20/1).

^1^H NMR (400 MHz, CDCl_3_) δ 7.10–6.98 (m, 90 H), 6.94–6.92 (m, 12 H), 6.74–6.71 (m, 12 H), 5.07–5.02 (m, 12 H), 4.57–4.25 (m, 12 H), 3.50–3.21 (m, 4 H), 2.90–2.75 (m, 24 H), 2.42 (m, 3 H), 2.13–2.11 (m, 3 H), 1.68–1.61 (m, 32 H), 1.42 (s, 9 H).

HRMS *m/z* (ESI) calcd. for C_221_H_214_N_24_O_9_S_6_Na_3_ (M+3Na)^3+^ 1201.0248, found 1201.0289.

### 3.5. Molecular Dynamics Simulation

The spatial structure of **7c** was obtained by performing molecular dynamics (MD) using the Amber99SB force field of the Gromacs (v2020.6). The solvent effect of water was simulated using the SPCE model implemented in the program. A cluster of 1000 structures was obtained by MD calculations performed at 300 K for 1000 ps after an equilibration time of 1000 ps. The cluster was analyzed with the gmx cluster tool to give an average structure.

## 4. Conclusions

In summary, in this work demonstrated the facile and efficient synthesis of fully substituted chiral triazole motifs by mild IrAAC and further assembly of them into stereocontrolled oligomers through metal-free iterative exponential growth strategies. Three oligotriazoles bearing different side chains were fabricated to illustrate the fidelity of this protocol, all of which were well identified by ^1^H NMR, MS, and SEC characterizations. Investigations of the photophysical performances of TPE-involved oligotriazoles as well as the related MD simulation results illustrated the potential application of this newly introduced structure in the development of advanced functional materials exhibiting chiroptical properties.

## Data Availability

The data presented in this study are available on request from the corresponding author.

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
