# Peer review of "Iridium-Catalyzed Synthesis of Chiral 1,2,3-Triazoles Units and Precise Construction of Stereocontrolled Oligomers"

_molecules, 2023, doi:10.3390/molecules28093726_

Round 1

Reviewer 1 Report

The authors reported an iridium-catalyzed synthesis of fully substituted 1,2,3-triazoles from proline-containing azides and S-containing alkynes. This protocol does not require harsh conditions and the selectivity is high. Using the synthesized triazoles as substrates, the authors constructed the stereocontrolled oligomers. The characterization of the monomers and oligomers has been almost completed. I would propose publication after minor corrections.

-        the abbreviations ISG (Scheme 1) and IEG (Scheme 2) should be clarified.

-        Compounds 3 should be added on synthetic schemes.

-        Scheme 2. Compound 5 should be checked (it seems that it should be OH instead of OTs).

-        The data of 13C NMR spectra should be added for compounds 1-8.

-        The isolated masses (in mg) and mmoles should be added for the synthesized compounds.

Author Response

The authors reported an iridium-catalyzed synthesis of fully substituted 1,2,3-triazoles from proline-containing azides and S-containing alkynes. This protocol does not require harsh conditions and the selectivity is high. Using the synthesized triazoles as substrates, the authors constructed the stereocontrolled oligomers. The characterization of the monomers and oligomers has been almost completed. I would propose publication after minor corrections.

- the abbreviations ISG (Scheme 1) and IEG (Scheme 2) should be clarified.

Response: thanks very much for the valuable suggestion. Related clarifications have been added in Scheme 1 and Scheme 2 accordingly.

- Compounds 3 should be added on synthetic schemes.

Response: thanks very much for the valuable suggestion. The structure of compound 3 has been added in Scheme 2A.

- Scheme 2. Compound 5 should be checked (it seems that it should be OH instead of OTs).

Response: we are sorry for the mistake. The structure of compound 5 in Scheme 2 has been corrected.

- The data of 13C NMR spectra should be added for compounds 1-8.

Response: thanks very much for the suggestion. Compound 2 is known, which has been well characterized in our previous research (Ref. 29 in the manuscript). The 13C NMR data and spectra of compounds 1, 3, 4 and dimer 5 have been added. Howbeit, due to the little amounts of other samples we left, we did not get efficient 13C NMR data of them. One thing to notice is that 13C NMR data are not usually provided for oligomers comprised of more than 3 monomers due to the superposition of signals.

- The isolated masses (in mg) and mmoles should be added for the synthesized compounds.

Response: thanks very much for the suggestion. Related experimental details have been added.

Reviewer 2 Report

The manuscript by Ding and col. describes iridium-catalyzed azide-thioalkyne cycloaddition reaction for the synthesis of chiral 1,2,3-triazoles that are next used for the stereocontolled preparation of oligomers.

This is an extension of the excellent work of the authors on iridium catalyzed 1,3 dipolar cycloaddition reactions, using in this case chiral azides. The obtained triazoles were further assembled into stereocontrolled oligotriazoles. The structure and uniformity of these oligomers  have been extensively characterized by the authors by NMR, size-exclusion chromatography and mass spectrometry, and their stereoregularity was studied through CD and circular polarized luminescence analysis.

The work is clearly presented and the conclusions are reasonably supported by the data. The precedents are well documented and the authors do not abuse on self-citation. I think that the authors have demonstrated the value of their methodology and they provide a nice contribution to the synthesis of triazole derivatives and metal catalyzed cycloaddition reactions.

Even though it is stated in the experimental that the internal standard for 13C NMR is CDCl3 at 77.0 ppm, I do not see 13C NMR peak list or spectra in ESI. I am surprised by the fact that the authors do not provide 13C NMR spectra, since the identity of new molecules should be established on the basis of their 1H and 13C NMR spectra and HRMS and evidences of their purity should be provided through image copies of both 1H and 13C NMR spectra or/and chromatograms.

The authors do not report the state of triazole molecules (solid, liquid). If solid, melting point should be provided.

In summary, I think that the authors report a nice piece of work that desires to be published in Molecules after the proper evidences of the identity and purity of new compounds are provided.

Minor issues

The graphical abstract is blurry (might be due to pdf production)

Scheme 2. The caption for part B “Assembly of stereocontroled oligotriazoles” seems confusing for me (I am not a native English though)

Author Response

The manuscript by Ding and col. describes iridium-catalyzed azide-thioalkyne cycloaddition reaction for the synthesis of chiral 1,2,3-triazoles that are next used for the stereocontrolled preparation of oligomers.

This is an extension of the excellent work of the authors on iridium catalyzed 1,3-dipolar cycloaddition reactions, using in this case chiral azides. The obtained triazoles were further assembled into stereocontrolled oligotriazoles. The structure and uniformity of these oligomers have been extensively characterized by the authors by NMR, size-exclusion chromatography and mass spectrometry, and their stereoregularity was studied through CD and circular polarized luminescence analysis.

The work is clearly presented and the conclusions are reasonably supported by the data. The precedents are well documented and the authors do not abuse on self-citation. I think that the authors have demonstrated the value of their methodology and they provide a nice contribution to the synthesis of triazole derivatives and metal catalyzed cycloaddition reactions.

Even though it is stated in the experimental that the internal standard for 13C NMR is CDCl3 at 77.0 ppm, I do not see 13C NMR peak list or spectra in ESI. I am surprised by the fact that the authors do not provide 13C NMR spectra, since the identity of new molecules should be established on the basis of their 1H and 13C NMR spectra and HRMS and evidences of their purity should be provided through image copies of both 1H and 13C NMR spectra or/and chromatograms.

Response: thanks very much for the suggestion. Compound 2 is known, which has been well characterized in our previous research (Ref. 29 in the manuscript). The 13C NMR data and spectra of compounds 1, 3, 4 and dimer 5 have been added. Howbeit, due to the little amounts of other samples we left, we did not get efficient 13C NMR data of them. One thing to notice is that 13C NMR data are not usually provided for oligomers comprised of more than 3 monomers due to the superposition of signals.

The authors do not report the state of triazole molecules (solid, liquid). If solid, melting point should be provided.

Response: thanks very much for the valuable suggestion. Related experimental details have been added.

In summary, I think that the authors report a nice piece of work that desires to be published in Molecules after the proper evidences of the identity and purity of new compounds are provided.

Minor issues

The graphical abstract is blurry (might be due to pdf production)

Response: thanks very much for the valuable comment. It has been replaced accordingly.

Scheme 2. The caption for part B “Assembly of stereocontrolled oligotriazoles” seems confusing for me (I am not a native English though).

Response: thanks very much for the valuable suggestion. The use of “assembly” in the manuscript has been adjusted accordingly.

Reviewer 3 Report

This paper was well-structured and well-written. The authors provided key findings and contributions of the research, including the use of IrAAC for the synthesis of fully substituted chiral triazole motifs and the fabrication of stereo controlled oligomers through metal-free exponential growth strategies. The authors provided additional detail on the experimental procedures and characterization techniques used, as well as the potential applications of the newly introduced structure. The authors also provided thorough characterization of the synthesized compounds enhances the credibility and impact of the research.

In terms of potential areas for improvement, it may be helpful to provide more detail on the specific functionalized thioalkynes used in the study and how they were selected. Additionally, more information on the specific metal-free exponential growth strategies used to assemble the chiral triazole motifs into oligomers could help readers better understand the methodology.

Overall, this paper was a well-executed study with significant potential for impact in the field of advanced functional materials.

There are 1 more question, also 1 minor revision about reference.

1.     How about using the CuAAC condition, would CuAAC achieves the same result?

2.     Line 76~77,

In parallel, the chiral organic azide 2 was prepared from commercially available N-Boc-L-prolinol (>99.9% ee) through a simple two-step synthetic protocol.

This should include the detailed protocol and reference.

Author Response

This paper was well-structured and well-written. The authors provided key findings and contributions of the research, including the use of IrAAC for the synthesis of fully substituted chiral triazole motifs and the fabrication of stereocontrolled oligomers through metal-free exponential growth strategies. The authors provided additional detail on the experimental procedures and characterization techniques used, as well as the potential applications of the newly introduced structure. The authors also provided thorough characterization of the synthesized compounds enhances the credibility and impact of the research.

In terms of potential areas for improvement, it may be helpful to provide more detail on the specific functionalized thioalkynes used in the study and how they were selected. Additionally, more information on the specific metal-free exponential growth strategies used to assemble the chiral triazole motifs into oligomers could help readers better understand the methodology.

Response: thanks very much for the valuable suggestion. Related demonstrations have been added in the revised manuscript.

Overall, this paper was a well-executed study with significant potential for impact in the field of advanced functional materials.

There are 1 more question, also 1 minor revision about reference.

  1. How about using the CuAAC condition, would CuAAC achieves the same result?

Response: thanks very much for the valuable question. CuAAC has been used in the construction of sequence-defined polymers through iterative sequential or exponential growth strategies, which has been well documented in one review from our group (Polym. Chem., 2021, 12, 2668). Howbeit, only 1,4-disubstituted triazole motifs were generated in these cases, the function of which is limited as linkage. IrAAC could afford 1,4,5-trisubstituted triazole compounds with regiospecificity, thus enabling it an ideal tool for the precise construction of macromolecules by using the triazole skeleton as linkage and the carrier of side group at the same time, which has been achieved in our previous and this research.

  1. Line 76~77, In parallel, the chiral organic azide 2 was prepared from commercially available N-Boc-L-prolinol (>99.9% ee) through a simple two-step synthetic protocol. This should include the detailed protocol and reference.

Response: thanks very much for the valuable suggestion. The synthesis and characterization of the chiral organic azide 2 has been illustrated in detail in our previous work (Ref. 29). Related demonstrations and citation have been added accordingly.

Round 2

Reviewer 1 Report

The authors have responded all reviewer's comments.